# Assessment of the Capability of Landsat and BiodivMapR to Track the Change of Alpha Diversity in Dryland Disturbed by Mining

**Yan Zhang [1], Jiajia Tang [2], Qinyu Wu [1,*], Shuai Huang [2], Xijun Yao [2,3] and Jing Dong [2]**

[1] School of Public Policy & Management, China University of Mining and Technology, Xuzhou 221008, China
[2] Engineering Research Center of Ministry of Education for Mine Ecological Restoration, School of Environment and Spatial Informatics, China University of Mining and Technology, Xuzhou 221008, China
[3] Institute of Territorial and Spatial Planning of Inner Mongolia, Hohhot 010070, China
* Correspondence: qywu@cumt.edu.cn; Tel.: +86-18051379737

**Abstract:** Remotely sensed spectral diversity is a promising method for investigating biodiversity. However, studies designed to assess the effectiveness of tracking changes in diversity using historical satellite imagery are lacking. This study employs open-access multispectral Landsat imagery and the BiodivMapR package to estimate the multi-temporal alpha diversity in drylands affected by mining. Multi-temporal parameters of alpha diversity were identified, such as vegetation indices, buffer zone size, and the number of clusters. Variations in alpha diversity were compared for various plant communities over time. The results showed that this method could effectively assess the alpha diversity of vegetation ($R^2$, 0.68). The optimal parameters used to maximize the accuracy of alpha diversity were NDVI threshold, 0.01; size of buffer zones, 120 m × 120 m; number of clusters, 100. The root mean square error of the alpha diversity of herbs was lowest (0.26), while those of shrub and tree communities were higher (0.34–0.41). During the period 1990–2020, the study area showed an overall trend of increasing diversity, with surface mining causing a significant decrease in diversity when compared with underground mining. This illustrates that the quick development of remote sensing and image processing techniques offers new opportunities for monitoring diversity in both single and multiple time phases. Researchers should consider the plant community types involved and select locally suitable parameters. In the future, the generation of long-time series and finer resolution maps of diversity should be studied further in the aspects of spatial, functional, taxonomic, and phylogenetic diversity.

**Keywords:** environmental assessment; alpha diversity; remote sensing; mining; vegetation; biodivMapR

## 1. Introduction

A total of 41% of the Earth's land is dryland, and 38% of the earth's population lives in dryland areas [1]. Nonetheless, ecosystems in dryland provide a lot of important ecosystem services, such as primary production, food supply, nutrient circulation, water and soil conservation, habitat, and leisure and recreational opportunities [2,3]. Moreover, biodiversity in drylands is more sensitive to climate change and other disturbances. In recent years, drylands have been increasingly disturbed by activities such as urbanization, mining, and high-intensive agriculture. These disturbances cause vegetation removal, water consumption, and various types of damage to landscapes, and, consequently, threaten biodiversity [4,5]. This requires timely monitoring of biodiversity to provide scientific data in support of land degradation assessment and ecological restoration [5]. Currently, many studies have conducted rapid and large-scale monitoring of land use change, total primary productivity, vegetation coverage, surface water, soil, and so on in drylands, but the monitoring of biodiversity still lacks adequate attention [6].

Traditionally, the monitoring of biodiversity has been conducted using field surveys organized by creating and monitoring a sampling quadrat. This kind of field survey usually covers only a small area, but its disadvantages are high cost, high time-consumption, and sampling and identification biases [7]. With the rapid development of remote sensing and image processing technology, more and more remote sensing images are freely available to use for continuous, large-scale, and remote environmental monitoring. Alpha diversity, a basic index for biodiversity, indicates the diversity that exists within a typical area or ecosystem. It is usually measured by the number of species, or the richness of species, within that ecosystem [8]. Alpha diversity has drawn much attention, given that it has a strong relationship with ecosystem productivity and resilience. Generally, the diversity index based on a field survey includes species richness [9], the Shannon–Wiener index [10], and the Simpson index. References [7,11] proposed a spectral variation hypothesis (SVH) as a new way to estimate biodiversity from remotely sensed images. According to SVH theory, remotely sensed images can reflect the spatial variation of the environment, because they have strong spectral and spatial heterogeneity. This spatial variability is related to species diversity. Subsequently, SVH theory was adopted by many studies in several ecosystems to rapidly investigate diversity [12–14]. Based on SVH theory, Féret and de Boissieu [15] developed the BiodivMapR package in R software to map alpha and beta diversity from imaging spectroscopy. The BiodivMapR package was successfully applied to the Amazonian forest [16], temperate mixed forest [17], a savanna [18], and cultivated areas [19]. It can be seen that remote sensing is becoming a potential approach to address the challenge of diversity observation across large spatial and time scales [20,21].

Notable progress has been made in the method of estimating diversity indices using multispectral and hyperspectral imagery. As investigated by Kacic and Kuenzer [22], the current study on the observation of diversity, based on remote sensing, presents a strong focus on mono-temporal resolution. Few studies have paid attention to multi-temporal monitoring. Obviously, mono-temporal observation could provide information on the change in diversity index, but this limits the detection of a change in diversity and estimation of a loss or gain [23–25]. This is particularly the case for ecosystems in drylands under pressure from disturbances, such as mining, erosion, and pollution. As time goes by, a large number of remote sensing images have been acquired and archived. For example, Landsat data have more than a 50-year history [26,27]. This provides an opportunity for the multi-temporal monitoring of diversity. However, studies that discuss the feasibility of multi-temporal monitoring of diversity using remotely sensed imagery are lacking.

Therefore, we assessed the capability of Landsat and BiodivMapR to track the change in alpha diversity in drylands under mining disturbance. The main purposes were the following: (1) to identify the optimal parameters of Landsat and BiodivMapR in the study area that can be used for monitoring diversity; (2) to investigate the relationship of estimated alpha diversity and field surveyed data related to the Shannon–Wiener index; (3) to discuss the effects of vegetation community types on the accuracy of diversity data; and (4) to detect the change of alpha diversity in the study area. This study aims at providing scientific and technical support for environmental impact assessment and management in drylands under mining disturbance.

## 2. Materials and Methods

### 2.1. Study Area

The study area was more than 908 km$^2$ and located in the east of the dryland in Asia, in northern China (38°52′–39°41′ N, 109°51′–110°46′ E; Figure 1). The local average annual rainfall is 200 to 500 mm, and the evaporation is 2000 to 3000 mm [4]. This shows that the study area is a typical dryland environment. The sandy soil supports sparse vegetation, while a river runs through the middle of the study area. The northern part of the study area is aeolian sand landform, and the south is loess hilly landform [4]. The study area is rich in coal resources. More than 52.39% of the study area has been disturbed by underground mining or surface mining since the 1990s (Figure 1).

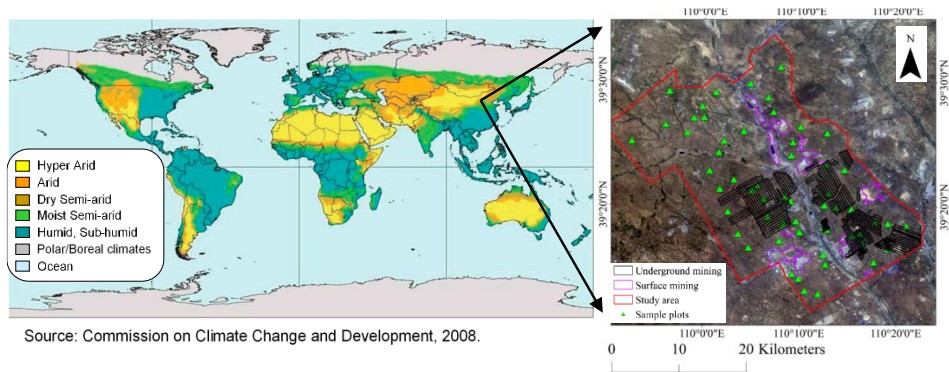

**Figure 1.** The geographical location of the study area.

*2.2. Data*

2.2.1. Remote Sensing

This study used four Landsat images, acquired in 1990, 2000, 2010, and 2020, to extract plant alpha diversity data. The images were recorded by the thematic mapper sensor on Landsat 5 satellite, except for the 2020 image from Landsat 8/OLI sensor, considering that the Landsat5 satellite was decommissioned in 2013 (Table 1). The images were selected from the most mature month of vegetation to avoid impact from phenology and other factors and were corrected for atmospheric and geometric issues with a spatial resolution of 30 × 30 m [28].

**Table 1.** Information for the employed Landsat imagery used in this study.

| Name | Date Acquired | Cloud Coefficient |
|------|---------------|-------------------|
| LT51270331990241BJC00 | 19900829 | 0.3% |
| LT51270332000237BJC00 | 20000824 | 0.1% |
| LT51270332010258IKR00 | 20100915 | 0.2% |
| LE71270332020252EDC00 | 20200908 | 0.1% |

2.2.2. Field Surveys

In September 2022, a time of peak plant diversity, 63 sampling plots were randomly established in the entire study area (Figure 1). Four quadrats were established in each plot, for a total of 252 quadrats that were investigated. For each quadrat, the species, numbers of individuals of each species, and growth status of the plants were surveyed. The Shannon–Wiener index was determined using the field data and Microsoft Excel [10], which accounted for the number of species and their relative evenness. The Shannon–Wiener index ($H'$) varied from 0 in plots with one dominant species to an undetermined maximum in plots with equally abundant species:

$$H' = -\sum_{i=1}^{N} p_i \ln(p_i) \tag{1}$$

where $N$ is the total number of species; $p_i$ is the abundance value of $i$th species.

*2.3. Methods*

2.3.1. Estimation of Alpha Diversity

The present study estimated alpha diversity based on Landsat images using the package BiodivMapR [15] running in the R Environment [29]. The method can be described in three steps (Figure 2).

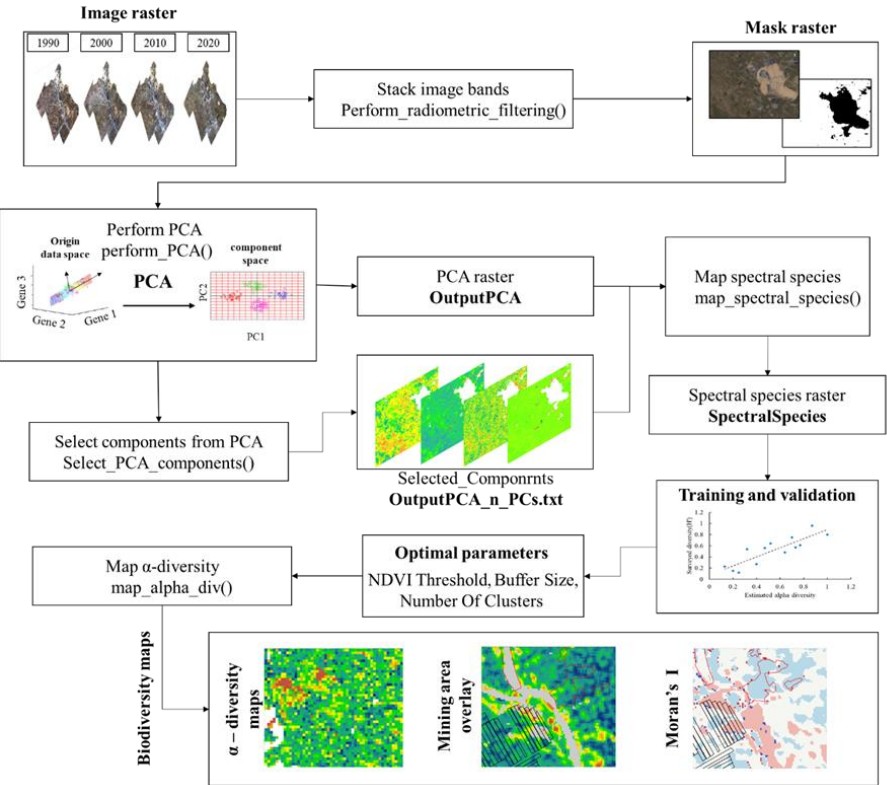

**Figure 2.** Diagrammatic representation of the steps used to estimate alpha diversity.

The steps include the following: (1) Image preprocessing was applied to the spectral data to mask clouds, shade, and non-vegetated pixels. To do this, a thresholding process was applied to the blue band, near-infrared band and Normalized Digital Vegetation Index (NDVI, range from −1 to 1) of Landsat images, respectively [30]. In order to reduce the multiplicative factors caused by light conditions, the spectral data was processed by continuum removal transformation to highlight the absorption and reflection characteristics of the spectrum and to normalize them to a uniform spectral background; (2) Principal component analysis (PCA) was then performed to decrease the spectral dimensionality, and the first four principal components, including most of the vegetation information, were selected to form new stack images. A k-means clustering algorithm was used to categorize image pixels within a user-defined number of clusters of spectral species, and the total number of spectral species defined within the scene. Here, BiodivMapR assumes that the spectral space corresponding to a landscape is a combination of subspaces, each of them related to one or several species sharing similar spectral signatures. Each of these subspaces can be referred to as a "spectral species" [15]. (3) A distribution pattern of spectral species was obtained for the scene, and the alpha diversity was computed from this distribution.

In the above process, NDVI threshold, the number of clusters and the total number of spectral species are important calculation parameters [15]. To identify the optimal NDVI threshold, a buffer zone of 60 × 60 m was set around each quadrat, and the number of clusters was set to 50. This study tested the NDVI threshold between 0 and 0.05 with 0.01 steps to mask non-vegetated pixels. Based on the optimal NDVI threshold, we compared the overall accuracy of buffers of 60 × 60 m, 90 × 90 m, 120 × 120 m, and 150 × 150 m around each quadrat, which corresponded to areas of 2 × 2, 3 × 3, 4 × 4, and 5 × 5 pixels in Landsat images. For the total number of spectral species, we tested 10, 20, 50, 100, and 150 spectral species during the process of principal component analysis and k-means clustering, and then compared the overall accuracy.

2.3.2. Accuracy Assessment

In order to evaluate the accuracy, we divided the samples from 252 quadrats into two groups, one of which was used as training data (140 samples) to estimate alpha diversity, and the other was used as validation data (112 samples). The training data was used to determine the best calculation parameters. After estimating the alpha diversity, we used the fieldd surveyed Shannon–Wiener index from the validation group and the estimated alpha diversity extracted from the diversity map to calculate the coefficient of determination ($R^2$) and root mean square error (*RMSE*). These two indicators were used to evaluate the accuracy of the estimation of alpha diversity [31,32]. Equations (2) and (3) were used to calculate $R^2$ and *RMSE*, respectively, as follows:

$$R^2 = 1 - \sum_{i=1}^{n}(\hat{y}_i - y_i)^2 / \sum_{i=1}^{n}(y_i - \overline{y})^2 \tag{2}$$

$$RMSE = \sqrt{\frac{1}{n}\sum_{i=1}^{n}(y_i - \hat{y}_i)^2} \tag{3}$$

where $y_i$ and $\hat{y}_i$ are the observations and estimations of the test dataset, respectively, $\overline{y}$ is the average of the observations, and $n$ is the number of observations of the test dataset.

To investigate the effects of BiodivMapR parameters and plant community types on the accuracy of the results, we calculated the coefficient of determination under different NDVI thresholds, buffer sizes, numbers of clusters, and community types, including pure tree, mixed forest, pure shrubs, mixed shrubs, and herbs.

2.3.3. Change Detection

After mapping the alpha diversity in 1990, 2000, 2010, and 2020, the imagery difference method was used to detect the changes for the periods 1990–2020, 1990–2000, 2000–2010, and 2010–2020. Hot spot analysis in ArcGIS 10.5 (ESRI, Redlands, CA, USA) was performed to identify the spatial clustering location of high or low values of alpha diversity. In addition, we extracted the estimated alpha diversity values of typical plant communities in 1990, 2000, 2010, and 2020, to observe the dynamic changes in plant diversity over time.

## 3. Results

### 3.1. The Effect of Parameters on Accuracy

Alpha diversity computed by the Landsat image was positively associated with the field surveyed Shannon–Wiener index, but the accuracy of the estimation of alpha diversity differed among calculation parameters (Figure 3).

The study found a positive correlation between alpha diversity calculated from Landsat images and field-surveyed data. The best results were obtained using an NDVI threshold of 0.01, a buffer zone of 120 × 120m and clusters of 50 (Figure 3).

According to the effect of BiodivMapR parameters on the accuracy, we estimated alpha diversity for 1990, 2000, 2010, and 2020 with the parameter values at the highest coefficient of determination. The highest value of correlation coefficients of alpha diversity and the Shannon–Wiener index was 0.6827 (Table 2).

**Table 2.** Summary of linear regressions for Shannon–Wiener index with spectral diversity.

| Estimated Value | Surveyed Value | Correlation Coefficient ($R^2$) | *p*-Value | *RMSE* |
|---|---|---|---|---|
| Alpha diversity | Shannon–Wiener index | 0.6827 | 0.002 | 5.68 |

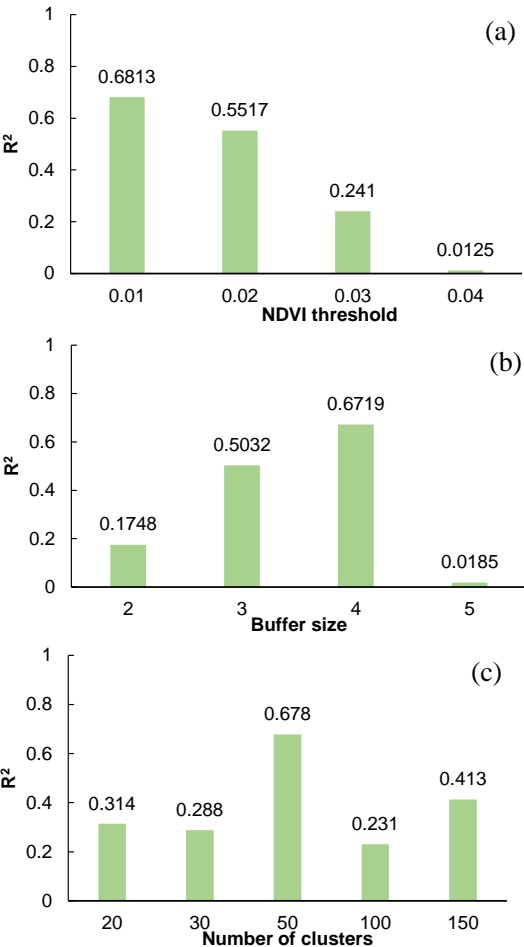

**Figure 3.** The relationship of key parameters with the coefficient of determination, (**a**) NDVI threshold, (**b**) Buffer size, (**c**) Number of clusters.

### 3.2. The Effect of Plant Community Types on Accuracy

In this study, we compared the accuracy of the estimation of alpha diversity for different community types, which were classified as pure tree stands, mixed forest stands, pure shrubs, and mixed shrubs and herbs (Figure 4).

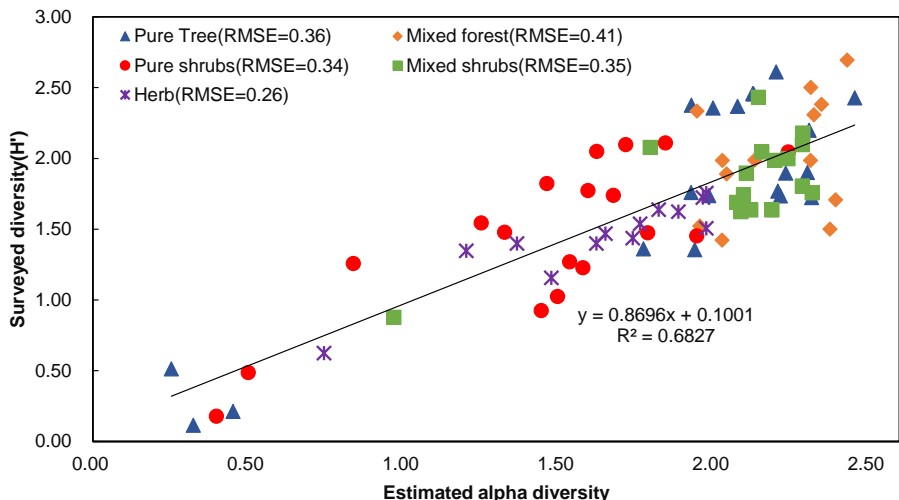

**Figure 4.** The spectral diversity of different vegetation community types in 2020.

Among the community types, herbs had the lowest fitting error (RMSE = 0.26), while the values of alpha diversity ranged from 1.2 to 2.0, indicating that the spectral diversity of herbs was more homogeneous than the spectral diversity of other community types analyzed here. The Landsat imagery and BiodivMapR package in R software had a better capability to reflect the diversity of homogeneous herbs, while the fitting errors of trees (RMSE = 0.38) and shrubs (RMSE = 0.34) were not as good. For trees, the RMSE between estimated alpha diversity and surveyed diversity was 0.36 for the pure tree community and 0.41 for mixed forest. The values of estimated alpha diversity for mixed forests were generally high, indicating that the level of diversity was higher and more variable for mixed forests than for pure tree stands. The RMSEs of mixed shrubs and pure shrubs were very similar, and the estimated alpha diversity values of pure shrubs were in a broad range from 0.4 to 2.0, while the values of mixed shrubs were concentrated between 2.0 and 2.5; this indicated the higher richness of the shrub community in the study area and the significant differences in diversity levels.

### 3.3. Spatial and Temporal Dynamics of Alpha Diversity

Figure 5 shows the alpha diversity of the study area in 1990, 2000, 2010, and 2020. The values of alpha diversity in 2020 were concentrated in the range of 0.24–2.42, with a mean and standard deviation of 2.00 and 0.17, respectively. This was in contrast to 2010, showing an increase of 0.08 in the mean and a decrease of 0.23 in the standard deviation. Meanwhile, in 2000, the mean and standard deviation were 1.83 and 0.23, respectively, and in 1990, they were 1.90 and 0.23, respectively.

During the 30 years from 1990 to 2020, the spatial extent of the area where alpha diversity increased was 567.95 km$^2$, accounting for 62.55% of the study area. Spatially, the areas with an increase in diversity were mainly located in the northwestern and central parts of the study area, with a significant increase in alpha diversity in the desert areas situated in the northeastern region and along both sides of the river, as well as in a small area overlapping with part of the mining area. The increase in diversity is mainly caused by many mine restoration projects including reforestation, land reclamation, water and soil conservation. The area of decreased diversity covered 340.02 km$^2$, accounting for 37.45% of the total area and was spatially distributed in the southeastern part of the study area, especially in the area where surface mines exist.

From 1990 and 2000, the study area exhibited an overall decrease in alpha diversity, with the proportions of the area showing increasing and decreasing alpha diversity covering 38.17% and 61.78% of the area, respectively (Figure 6). However, the variation of alpha diversity during this period was small and in the range of −0.7 to 0.7. From 2000 to 2010, 618.55 km$^2$ or 68.12% of the study area experienced an increase in alpha diversity, while 289.43 km$^2$ or 31.88% experienced a decrease. Spatially, the areas of increased alpha diversity were mainly located in the northern and western parts of the study area, while the areas with decreased alpha diversity were located in the southeastern and central parts. A small number of mining areas and urban construction areas overlapped with the areas showing a significant decrease, and along both sides of the river in the north overlapped with the areas experiencing a significant increase. From 2010 to 2020, 524.79 km$^2$ or 57.8% experienced an increase in alpha diversity, and 383.18 km$^2$ or 42.2% of the total area experienced a decrease. Spatially, the increase in diversity mainly occurred in the northwestern and central parts of the study area, and the areas with a significant decrease overlapped with the mining area to a high degree.

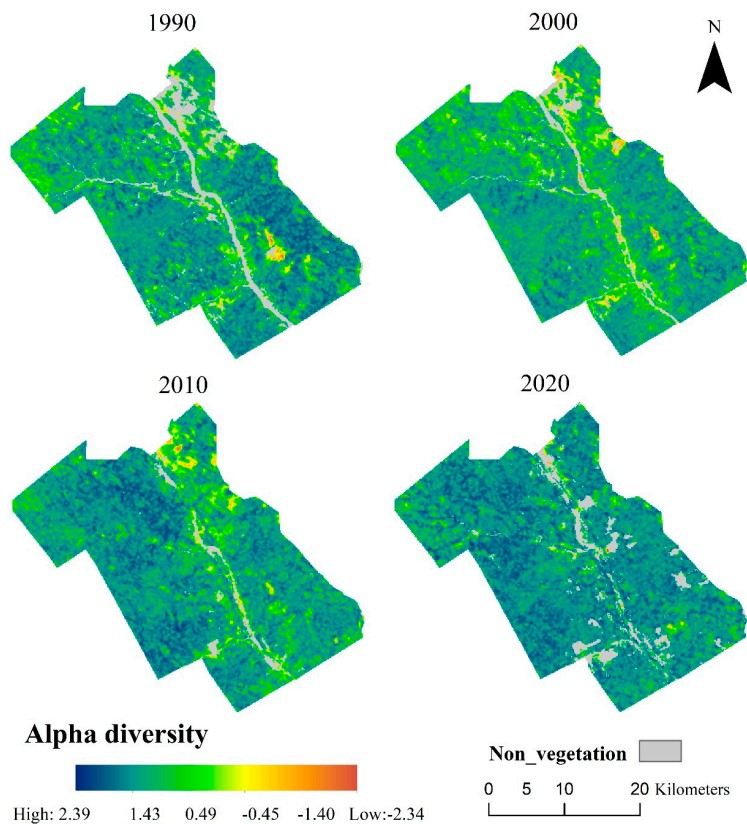

**Figure 5.** Alpha diversity of the study area in 1990, 2000, 2010, and 2020.

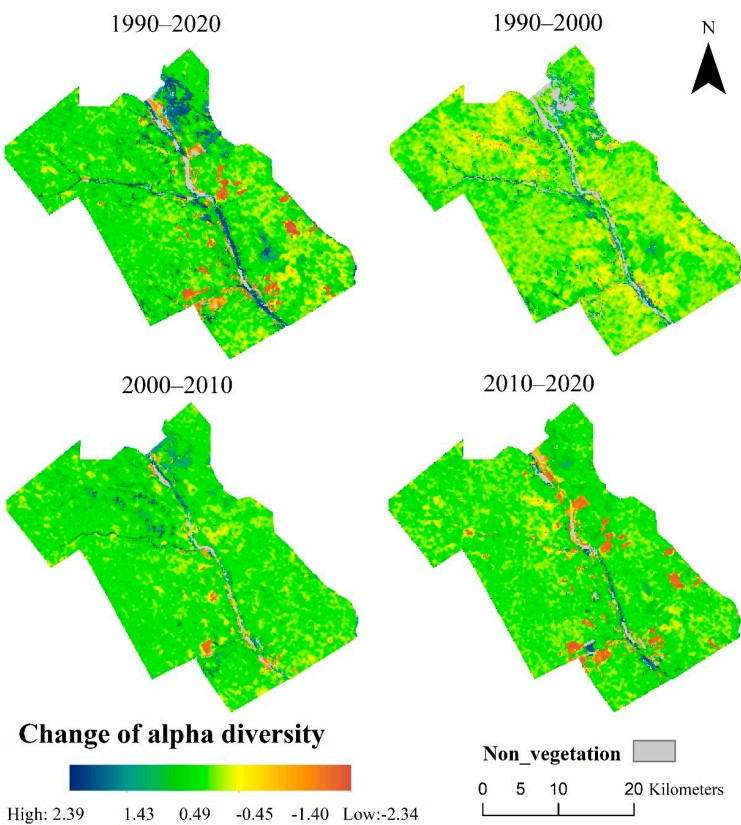

**Figure 6.** Changes in alpha diversity in the study area from 2000 to 2020.

In summary, diversity fluctuated over the 30 years of the present study, with the 10 years from 2000 to 2010 showing a significant increase in diversity, contributing most to the eventual increase in diversity levels, and the 10 years from 2010 to 2020 showing some areas with a significant decline in alpha diversity caused by coal mining, which had a significant impact on the decrease in the levels of diversity.

### 3.4. Hot and Cold Spots of Diversity Change in the Study Area

Hot spot analysis was performed based on the map of the change in alpha diversity from 1990 to 2020. Figure 7 shows the hot and cold spot areas of the estimated variation in alpha diversity for the study area. The hot and cold spot areas stand for the spatial clustering locations of increases and decreases in alpha diversity from 1990 to 2020. The area of the cold spots accounted for 21.07% (99%, 95%, and 90% confidence accounted for 1.10%, 5.56%, and 4.41%, respectively) and that of hot spots accounted for 20.82% of the study area (99%, 95%, and 90% confidence accounted for 14.59%, 3.72%, and 2.50%, respectively).

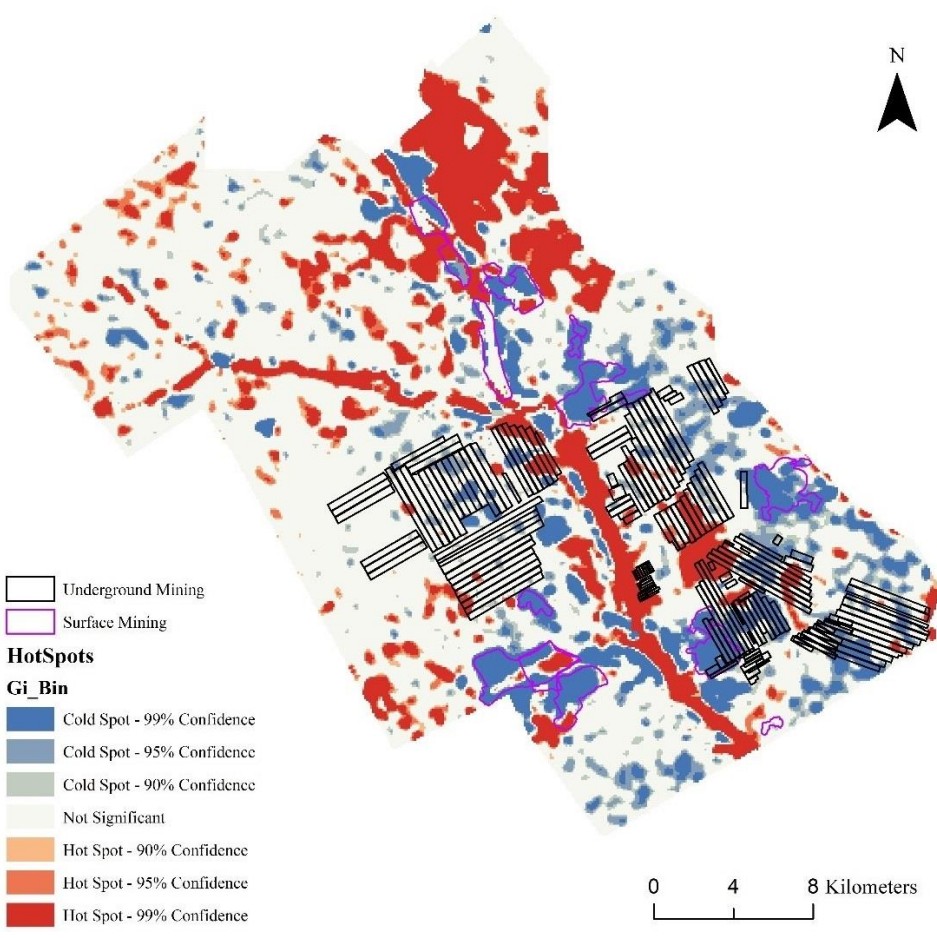

**Figure 7.** Cold and hot spot areas of changes in alpha diversity from 1990 to 2020.

Hot spot areas of alpha diversity were mainly located along the rivers in the center of the study area and around the sand dunes in the north, with some smaller hot spots scattered over the entire region. This pattern of hot spots was mainly attributed to the control of desertification and green engineering. Within the areas with underground and surface mining, only a few areas were hot spots with an increase in alpha diversity. The cold spots were mainly located in the urban construction areas around the river and the surface mining areas. Little overlap was observed between underground mining areas and cold spots.

### 3.5. Changes in Alpha Diversity of Different Plant Communities

The changes in alpha diversity among the 11 typical plant communities in the study area are presented in Figure 8. Populations of herbs, such as Pennisetum centrasiaticum and Cleistogenes squarrosa, declined significantly from 1990 to 2000, but this situation improved after 2000, indicating an improvement in the environment of the study area.

As for shrubs, the alpha diversity of Hippophae rhamnoides increased insignificantly from 2.11 in 2010 to 2.21 in 2020. The alpha diversity of Artemisia ordosica and Caragana korshinskii fluctuated, with significant differences ($p < 0.05$), 1.79 in 2000 and 1.98 in 2020. In addition, no significant difference was observed in the change in diversity for Salix mongolica. This suggests that Artemisia ordosica and Caragana korshinskii, as pioneer species used during ecological restoration, played a significant role in enhancing diversity, but the enhancement effect of Spiraea mongolica was relatively poor.

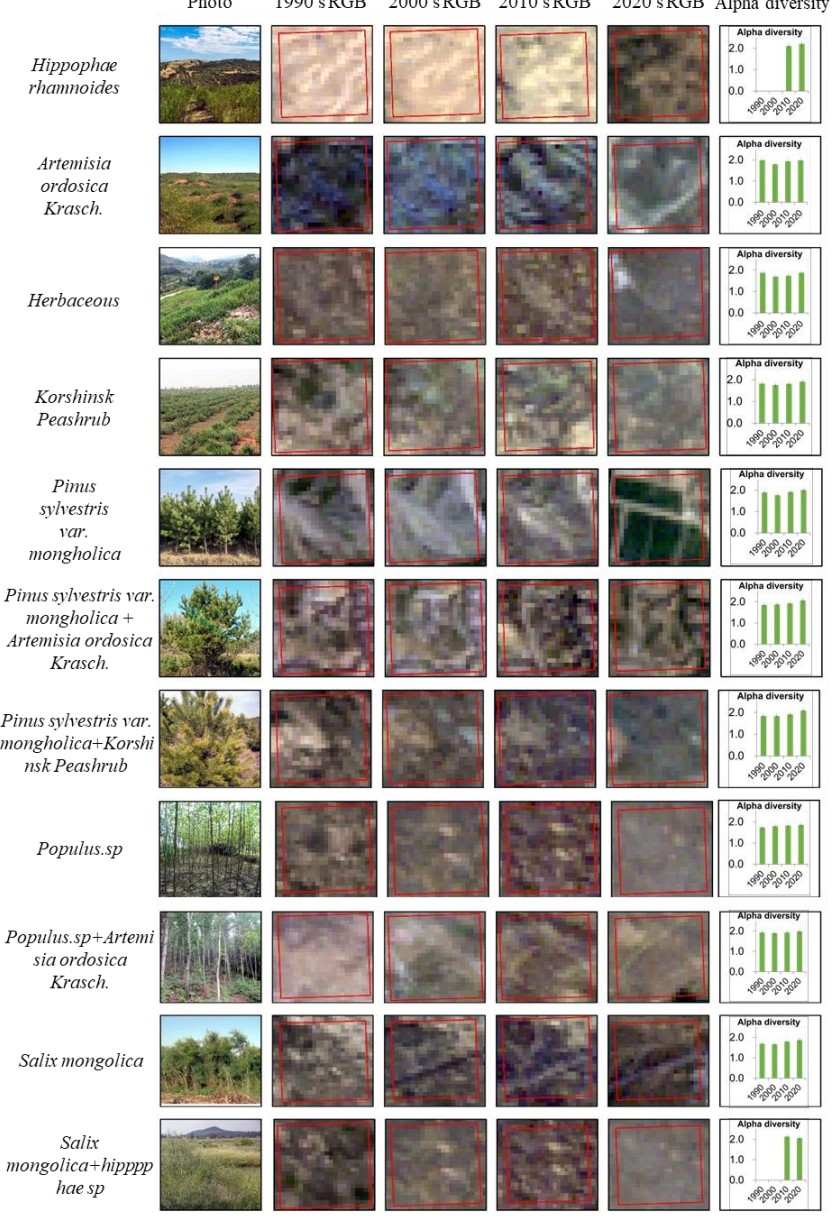

**Figure 8.** Changes in alpha diversity among different plant communities. No data were available for the Hippophae rhamnoides and Salix mongolica + Hippophae communities in 1990 and 2000 because they were introduced into the study area during vegetation restoration efforts after 2000.

Among the trees, the diversity of pure plantation forests of Pinus sylvestris var. mongolica declined from 1990 to 2000 and increased after 2000, while that of the natural mixed Pinus sylvestris var. mongolica forests continued to rise steadily from 1990. By contrast, the diversity levels of pure and mixed Populus forests were relatively stable over the entire study period. This indicates that planting Pinus sylvestris var. mongolica, which grows rapidly, can help in increasing levels of diversity in a short period of time but with instability, while Populus has a long growing period and diversity in stands with this species increases more slowly (Figure 8).

## 4. Discussion

### 4.1. Implications for Plant Alpha Diversity Monitoring

Drylands typically have sparse vegetation and low diversity, creating a challenge to monitoring species diversity over space and time. Current efforts to monitor the environment with remote sensing in arid areas have given little attention to diversity. A few studies have used surrogate variables, such as the NDVI and its derivatives, for estimating biodiversity [33–37]. In recent years, the method based on the spectral variation hypothesis for diversity monitoring has gradually gained attention [38]. The present study shows that the use of Landsat and BiodivMapR can help researchers realize an accurate estimation of alpha diversity. A buffer size of four times the Landsat image pixel (120 m × 120 m) is suitable for accurate diversity estimation in low vegetation, heterogeneous areas. Accuracy of diversity estimation in lush vegetation areas may be low due to the small difference between pixel texture and spectral characteristics. Community types impact accuracy, with herb habitats having the smallest error and mixed forests the poorest estimate (Figure 4). Separate estimation of different community types can improve accuracy [12,13].

The present study shows that using Landsat imagery and the spectral variation hypothesis is effective in tracking change in diversity. The advantage of this method is that it can quickly identify regions with a loss or gain in diversity. Thus, this method can provide reliable spatial data for the assessment of land degradation and ecological restoration. However, the coefficient of determination only reaches 0.68. This is mainly caused by the fact that the Landsat images have a moderate spatial resolution (30 m × 30 m) and spectral resolution (7 wave bands for Landsat 5, 9 wave bands for Landsat 8). Landsat images can be used for distinguishing different plant life forms, such as trees, shrubs and herbs. However, it is difficult to use Landsat images to identify plant species one by one. In order to evaluate the diversity of local scales more accurately, hyperspectral data should be considered [20]. In addition, this study recognizes that the diversity of some communities remains stable over time, but the diversity index of most plant communities changes annually as a result of ecological succession, human interference, climate fluctuations, and other factors [4,39]. To quantitatively describe the effects of these factors on the inter-annual variation of the diversity index, it is necessary to, as far as possible, select images with the same imaging season and time of day and correct the images to eliminate the seasonal fluctuation.

Drylands have been increasingly affected by human activity over time. For example, mining activities have shifted from humid to remote arid areas in the past 20 years [5,40,41]. Environmental impact assessments previously focused on land use and vegetation changes but modern restoration efforts aim to restore habitat structure and function [42–45].

Diversity is an important indicator of the structure and function of mine ecosystems [46]. In this study area, the stripping of surface vegetation by surface mining directly led to a reduction in alpha diversity, over and above the impact of underground mining, occurring more slowly. The present study shows that diversity monitoring with the help of remote sensing data provides a new perspective on impact assessments in mining areas and can be used as a new indicator. It should be emphasized that remote sensing data should be used to complement, not replace, in situ data on biological diversity.

*4.2. Research Limitations and Future Work*

There are a few limitations when using Landsat imagery and BiodivMapR to reveal vegetation diversity that need to be recognized. First, optimization of the BiodivMapR calculation parameters needs further enhancement. In this study, the detected diversity values between remote and in situ surveys had the highest accuracy values when using an NDVI threshold of 0.01, a 120 × 120 m buffer, and 50 clusters to calculate alpha diversity. At the NDVI threshold of 0.01 it is easy to recognize non-vegetation pixels from the image. However, plant community type, short-term precipitation, land use, and other factors can affect vegetation growth, so further calibration is required when applying remote sensing images acquired in different seasons or other timestamps in different spectral bands to calculate alpha diversity. Second, due to the long revisit period of Landsat and its optical imaging, there are less available Landsat images in some areas, which reduces the temporal and spatial resolution of diversity assessment. Third, the field survey data only covered 2020; this study lacked field survey data for accuracy assessment in past periods, including 1990, 2000, and 2010.

In view of these shortcomings, more research work needs to be completed. Remoted sensing based on unmanned aerial vehicles (UAVs) has developed rapidly in the past decade. UAV remote sensing has the advantages of low cost, being fast and easy to repeat, and is widely used in ecosystem monitoring [47,48]. In particular, the application of laser light detection and ranging (LiDAR) and hyperspectral imaging has further enhanced the capability of UAV remote sensing in diversity monitoring and ecosystem research [49,50]. In the future, UAV-based LiDAR and hyperspectral imaging could be used to estimate alpha diversity with finer spatial and temporal resolutions. Moreover, using temporal remote sensing data and spatiotemporal big data computing tools, such as the Google Earth Engine, to generate regional or global diversity sequence maps is a potential direction for future research [22,51,52]. This can provide rich data for monitoring and modeling global change. In addition to the alpha, beta, and gamma diversity indices from the spatial hierarchical perspective, other aspects of functional (structural, biophysical, and biochemical), taxonomic, phylogenetic, and genetic diversity deserve attention [53].

## 5. Conclusions

The goal of this study was to assess the capability of Landsat and BiodivMapR to track change in alpha diversity in dryland under mining disturbance. To achieve this objective, the alpha diversity in 1990, 2000, 2010, and 2020 was estimated. Then, the effects of calculation parameters and vegetation community types on the accuracy of the data were discussed.

The results showed that using Landsat imagery and the BiodivMapR package in R software can be effective in assessing alpha diversity. The overall correlation coefficient between alpha diversity and surveyed diversity was 0.6827 in the study area. The sensitive parameters included the NDVI threshold, buffer size, and the number of clusters analyzed. Among the community types, the smallest error in alpha diversity estimation was found in herb habitats, followed by shrub, and the poorest was in mixed forest. Based on the dynamics of multiple temporal maps of alpha diversity, hot and cold spots of diversity change can be easily spatially identified, to provide data useful for environmental assessment. It was found that although the study showed an overall increasing trend in diversity, surface mining, rather than underground mining, had caused a significant decrease in diversity.

The results indicate that Landsat data can be used to assess diversity and its changes in drylands, but the resolution of diversity maps is limited; multi-source data fusion and the production of a long-term time series of diversity maps should be considered in the future. There is a need to explore the uses of remote sensing to estimate other aspects of diversity, such as structural, biophysical, and biochemical diversity over space and time.

**Author Contributions:** Conceptualization, writing—original draft preparation, and writing—review and editing, Y.Z.; methodology, validation, formal analysis, and software, J.T. and J.D.; funding acquisition and resources, Q.W.; investigation, supervision, and project administration, X.Y. and S.H. All authors have read and agreed to the published version of the manuscript.

**Funding:** The work reported in this study was supported by the National Natural Science Foundation of China (Grant No. 51974313) and the Major Special Projects of the Third Comprehensive Scientific Exploration in Xinjiang (Grant No. 2022xjkk1005).

**Institutional Review Board Statement:** Not applicable.

**Informed Consent Statement:** Not applicable.

**Data Availability Statement:** The data presented in this study are available on request from the corresponding author.

**Acknowledgments:** We thank LetPub for its linguistic assistance during the preparation of this manuscript.

**Conflicts of Interest:** The authors declare no conflict of interest.

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
