# Peer review of "Assessment of the Capability of Landsat and BiodivMapR to Track the Change of Alpha Diversity in Dryland Disturbed by Mining"

_remotesensing, doi:10.3390/rs15061554_

Round 1

Reviewer 1 Report

Dear Author(s),

The paper tackles an exciting topic of assessing Landsat and BiodivMapR's potential for studying biodiversity variation in a study area located in China, between 1990 and 2020. The research is scientifically sound and may interest a lot of audiences. I like how you presented your work in a very straightforward, simple way. However, there are only a few remarks to address. Please check out my comments attached to the document

—Best of luck.

Author Response

The paper tackles an exciting topic of assessing Landsat and BiodivMapR's potential for studying biodiversity variation in a study area located in China, between 1990 and 2020. The research is scientifically sound and may interest a lot of audiences. I like how you presented your work in a very straightforward, simple way. However, there are only a few remarks to address. Please check out my comments attached to the document

—Best of luck.

Reponses 1: Thanks for your time. We have carefully read your comments on the attached .pdf file and revise the manuscript accordingly. The revision has been marked in red. Specifically, we would like report two major revisions to you.

One is Figure 1. We have replace the former map with a new one to show the global dryland and borders. This will help readers recognize the location of the study area

Another one is the comments of “What do you think can be done to improve these values (correlation coefficient R2) in future work? And what are the potential reasons for such moderately low values?”. We have added a brief discussion into chapter 4.1. It is “However, the coefficient of determination only reaches 0.68. This mainly caused by that the Landsat images have a moderate spatial resolution (30m × 30 m) and spectral resolution (7 wave bands for Landsat 5, 9 wave bands for Landsat 8). Landsat images can be used for distinguishing different plant life forms, such as trees, shrubs and herbs. However, it is difficult to use Landsat images to identify plant species one by one. In order to evaluate the diversity of local scales more accurately, hyperspectral data should be considered.”

Reviewer 2 Report

The manuscript Evaluation of Landsat and BiodivMapR's ability to track alpha diversity change in mining-disturbed arid areas by Zhang et al is well written.

The study used Landsat satellite imagery to assess alpha diversity in drylands affected by the mining and found that remote sensing and image processing can effectively track changes over time. Optimal parameters for accuracy were determined and the study area showed a trend of increasing diversity, with surface mining causing a decrease. However, there are too many details in the introduction. Please, shorten the paragraphs.

35-36, 44 must be moved to discussion or write a reference

Materials and Methods with Study Area - please move paragraph 102-105 at Discussion or a part of it, at least.

Next, Data with Remote sensing: This study used four Landsat images acquired in 1990, 2000, 2010, and 2020  to extract plant alpha diversity data. The images were recorded by the thematic mapper sensor on Landsat5 satellite except for the 2020 image from Landsat8/OLI sensor. The images were selected from the most mature month of vegetation to avoid impact from phenology and other factors and were corrected for atmospheric and geometric issues with a spatial resolution of 30x30 m, and  Field surveys.

Methods: alpha-diversity estimation is presented in detail the package BiodivMapR, Accuracy assessment and also Change detection.

Results present The effect of parameters on accuracy: The study found a positive correlation between alpha diversity calculated from Landsat images and field-surveyed data. The best results were obtained using a buffer zone of 120x120m and 50 clusters, with a coefficient of determination of 0.6827.

The effect of plant community types on accuracy: The study compared the accuracy of estimating alpha diversity of five community types: trees, mixed forests, pure shrubs, mixed shrubs and herbs. The best accuracy was found for homogeneous herbs, while accuracy was lower for trees and shrubs. The mixed forest stands had higher and more variable alpha diversity than pure tree stands, and mixed shrubs had higher richness and greater diversity than pure shrubs.  Spatial and temporal dynamics of alpha diversity: alpha diversity fluctuated over 30 years, with an increase from 2000 to 2010 and a decline from 2010 to 2020 due to coal mining. The decline had a significant impact on decreasing diversity levels.

However, not all results are relevant: there are many details and even some repetitions.

Next Hot and cold spots of diversity change in the study area: The study performed a hot spot analysis to identify areas of change in alpha diversity from 1990 to 2020. Hot spots were mainly located in riverside and sand dune areas, while cold spots were mostly in urban construction and surface mining areas. 

290-296 move at Discussions, there are no results but the author's opinion only.

Changes in alpha diversity of different plant communities: Herbs and shrubs showed changes in diversity from 1990 to 2020, with some species playing a significant role in increasing diversity. Tree species showed diverse results, with Pinus sylvestris var. mongolica showing instability, mixed P. sylvestris var. mongolica showing a steady increase, and Populus sp. showing stable diversity levels.

However, please keep the same species nomenclature (e.g. with/without the author's name)

Discussion with Implications for plant alpha diversity monitoring: The use of Landsat and BiodivMapR can accurately estimate alpha diversity in drylands, but the buffer zone size must be considered. A buffer size of four times the Landsat image pixel (120m x 120m) is suitable for accurate diversity estimation in low vegetation, heterogeneous areas. Accuracy of diversity estimation in lush vegetation areas may be low due to small difference between pixel texture and spectral characteristics. Community types impact accuracy, with herb habitats having the smallest error and mixed forests the poorest estimate. Separate estimation of different community types can improve accuracy. Human activities, such as mining, have impacted drylands. Environmental impact assessments previously focused on land use and vegetation changes but modern restoration efforts aim to restore habitat structure and function. Diversity monitoring with the help of remote sensing data provides a new perspective on impact assessments in mining areas and can be used as a new indicator. Remote sensing data should be used to complement, not replace, in situ data on biological diversity.

The conclusion section summarizes the main findings of the study and states the limitations of the study, as well as suggests future directions for research. It is concise, clear, and provides a clear answer to the research question posed in the introduction.

Author Response

The manuscript Evaluation of Landsat and BiodivMapR's ability to track alpha diversity change in mining-disturbed arid areas by Zhang et al is well written.

The study used Landsat satellite imagery to assess alpha diversity in drylands affected by the mining and found that remote sensing and image processing can effectively track changes over time. Optimal parameters for accuracy were determined and the study area showed a trend of increasing diversity, with surface mining causing a decrease. However, there are too many details in the introduction. Please, shorten the paragraphs.

Reponses 1: Thanks for carefully reviewing our work and providing very wonderful and concise statement.

35-36, 44 must be moved to discussion or write a reference.

Reponses 2: Agree, the mentioned sentence had been moved to discussion.

Materials and Methods with Study Area - please move paragraph 102-105 at Discussion or a part of it, at least.

Reponses 3: Agree, the mentioned sentence had been moved discussion.

Next, Data with Remote sensing: This study used four Landsat images acquired in 1990, 2000, 2010, and 2020  to extract plant alpha diversity data. The images were recorded by the thematic mapper sensor on Landsat5 satellite except for the 2020 image from Landsat8/OLI sensor. The images were selected from the most mature month of vegetation to avoid impact from phenology and other factors and were corrected for atmospheric and geometric issues with a spatial resolution of 30x30 m, and  Field surveys.

Reponses 4: We appreciate your help. This paragraph had been updated as you suggest.

Methods: alpha-diversity estimation is presented in detail the package BiodivMapR, Accuracy assessment and also Change detection.

Reponses 5: Thanks for your reminder. We agree that BiodivMapR has provide details. In order to ensure the integrity of this paper and provide readers with necessary information, we still briefly introduce the steps.

Results present The effect of parameters on accuracy: The study found a positive correlation between alpha diversity calculated from Landsat images and field-surveyed data. The best results were obtained using a buffer zone of 120x120m and 50 clusters, with a coefficient of determination of 0.6827.

Reponses 6: The mentioned paragraph had been shortened as you suggested.

The effect of plant community types on accuracy: The study compared the accuracy of estimating alpha diversity of five community types: trees, mixed forests, pure shrubs, mixed shrubs and herbs. The best accuracy was found for homogeneous herbs, while accuracy was lower for trees and shrubs. The mixed forest stands had higher and more variable alpha diversity than pure tree stands, and mixed shrubs had higher richness and greater diversity than pure shrubs. 

Reponses 7: Considering that the effect of plant community types on accuracy is quite important for reflect the capability of Landsat and BiodivMapR. We keep the detail description as it was. We hope you can understand.

Spatial and temporal dynamics of alpha diversity: alpha diversity fluctuated over 30 years, with an increase from 2000 to 2010 and a decline from 2010 to 2020 due to coal mining. The decline had a significant impact on decreasing diversity levels. However, not all results are relevant: there are many details and even some repetitions.

Reponses 8: The repetitions has been checked out and deleted.

Next Hot and cold spots of diversity change in the study area: The study performed a hot spot analysis to identify areas of change in alpha diversity from 1990 to 2020. Hot spots were mainly located in riverside and sand dune areas, while cold spots were mostly in urban construction and surface mining areas. 

290-296 move at Discussions, there are no results but the author's opinion only.

Reponses 9: The mentioned paragraph had been shortened as you suggested. The author's opinion had been moved to discussion.

Changes in alpha diversity of different plant communities: Herbs and shrubs showed changes in diversity from 1990 to 2020, with some species playing a significant role in increasing diversity. Tree species showed diverse results, with Pinus sylvestris var. mongolica showing instability, mixed P. sylvestris var. mongolica showing a steady increase, and Populus sp. showing stable diversity levels. However, please keep the same species nomenclature (e.g. with/without the author's name)

Reponses 10: The mentioned paragraph had been shortened as far as possible. The author's opinion had been moved to discussion. We keep the same species nomenclature. The author's name had been deleted.

Discussion with Implications for plant alpha diversity monitoring: The use of Landsat and BiodivMapR can accurately estimate alpha diversity in drylands, but the buffer zone size must be considered. A buffer size of four times the Landsat image pixel (120m x 120m) is suitable for accurate diversity estimation in low vegetation, heterogeneous areas. Accuracy of diversity estimation in lush vegetation areas may be low due to small difference between pixel texture and spectral characteristics. Community types impact accuracy, with herb habitats having the smallest error and mixed forests the poorest estimate. Separate estimation of different community types can improve accuracy. Human activities, such as mining, have impacted drylands. Environmental impact assessments previously focused on land use and vegetation changes but modern restoration efforts aim to restore habitat structure and function. Diversity monitoring with the help of remote sensing data provides a new perspective on impact assessments in mining areas and can be used as a new indicator. Remote sensing data should be used to complement, not replace, in situ data on biological diversity.

Reponses 11: Very wonderful and concise statement! Thank you for your help. The mentioned paragraph had been updated.

The conclusion section summarizes the main findings of the study and states the limitations of the study, as well as suggests future directions for research. It is concise, clear, and provides a clear answer to the research question posed in the introduction.

Reponses 12: Thanks for your comments.

Reviewer 3 Report

Dear authors,

After carefully reviewed your article “Assessment of the capability of Landsat and BiodivMapR to track the change of alpha diversity in dryland disturbed by mining” I found out that your work is interesting, and the presented methodology is clear and easy to follow. I also appreciated the fact that you emphasized the limitation of the methodology. However, there are a few things that may be better explained, se the specific comments below.

L36 - 37 not very clear here, are you compare the cold spot with the hot spots?

L 138 Figure 2 Would be better to add the training and validation part

L 141 reconsider band numbering as they are different for Landsat 5 and Landsat 8

L 151 this is rather a general comment concerning the methodology. You have to explain better what spectral species are and how you established the values for thresholding the blue, near infrared and NDVI bands.

L 154 I think deleting “pixels” is better to avoid any confusion

L 162 please add here that sample means the field derived values for alpha diversity from the 252 quadrats

L 163 - 164 It would be better to clarify the process of training and validation

L 206 Is this the spectral diversity, the same as species diversity from the methodological part?

L 381 You said that 50 clusters are producing the best results line 208.

Author Response

After carefully reviewed your article “Assessment of the capability of Landsat and BiodivMapR to track the change of alpha diversity in dryland disturbed by mining” I found out that your work is interesting, and the presented methodology is clear and easy to follow. I also appreciated the fact that you emphasized the limitation of the methodology. However, there are a few things that may be better explained, se the specific comments below.

Reponses 1: Thanks for your time.

L36 - 37 not very clear here, are you compare the cold spot with the hot spots?

Reponses 2: This is a wrong sentence, it had been deleted.

L 138 Figure 2 Would be better to add the training and validation part.

Reponses 3: Figure 2 had been revised as you suggest.

L 141 reconsider band numbering as they are different for Landsat 5 and Landsat 8

Reponses 4: Good reminder, the band number had been deleted.

L 151 this is rather a general comment concerning the methodology. You have to explain better what spectral species are and how you established the values for thresholding the blue, near infrared and NDVI bands.

Reponses 5: The definition of spectral species had been added. Considering that BiodivMapR package has provide the detail about the determination of the values for thresholding the blue, near infrared and NDVI bands, we add a reference rather than a lengthy explanation.

L 154 I think deleting “pixels” is better to avoid any confusion.

Reponses 6: Agree, deleted.

L 162 please add here that sample means the field derived values for alpha diversity from the 252 quadrats

Reponses 7: Agree, corrected.

L 163 - 164 It would be better to clarify the process of training and validation.

Reponses 8: A brief introduction had been added. It is “The training data was used for determine the best calculation parameters. After estimating the alpha diversity, we use the filed surveyed Shannon–Wiener index from the validation group and the estimated alpha diversity extracted from the diversity map to calculate the coefficient of determination (R2) and root mean square error (RMSE).”

L 206 Is this the spectral diversity, the same as species diversity from the methodological part?

Reponses 8: Sorry, this is a mistake. It should be alpha diversity. Corrected.

L 381 You said that 50 clusters are producing the best results line 208.

Reponses 9: Sorry, this is a mistake. Corrected.